# Bio-inspired Min-Nets Improve the Performance and Robustness of Deep Networks

**Philipp Grüning**
Institute for Neuro- and Bioinformatics
University of Lübeck
Lübeck, 23562
gruening@inb.uni-luebeck.de

**Erhardt Barth**
Institute for Neuro- and Bioinformatics
University of Lübeck
Lübeck, 23562
barth@inb.uni-luebeck.de

## Abstract

Min-Nets are inspired by end-stopped cortical cells with units that output the minimum of two learned filters. We insert such Min-units into state-of-the-art deep networks, such as the popular ResNet and DenseNet, and show that the resulting Min-Nets perform better on the Cifar-10 benchmark. Moreover, we show that Min-Nets are more robust against JPEG compression artifacts. We argue that the minimum operation is the simplest way of implementing an AND operation on pairs of filters and that such AND operations introduce a bias that is appropriate given the statistics of natural images.

## 1 Introduction

We present a novel network architecture that is based on the AND combination of pairs of linear filters. The linear filters in deep networks can be considered as basic measurements of the input signal, each filter capturing signal energy along some relevant direction in feature space. In differential geometry, the basic measurements are the main curvatures. In the case of 2-dimensional manifolds, it is known that the structure of the manifold is captured by the Gaussian curvature, which is the product of the main curvatures (do Carmo (1976)). In higher dimensions, the Riemann curvature tensor captures the structure of the manifold, and it is based on sums of products of main curvatures (OR combinations of pairwise AND combinations). Inspired by differential geometry, in computer vision, different algorithms have been designed to capture interest points in images, for example, by analysing the invariants and eigenvalues of the Hessian or of the structure tensor (Guiducci (1988); Jähne (1995)). Both the product and the minimum of the eigenvalues have been used for the detection of interest points (Shi & Tomasi (1994)).

In related work on feature product (FP)-nets (Grüning et al. (2021)), the AND operation has been implemented by using explicit multiplications. FP-nets have been shown to improve the performance of state-of-the-art networks. Moreover, FP-units are hyperselective (for more information on hyperselectivity, see (Paiton et al. (2020)) and (Vilankar & Field (2017))) and thus more robust against compression artifacts and adversarial attacks (Grüning et al. (2021)). As an alternative to the computationally more intensive multiplications, the log-nets introduced in Grüning et al. (2020) are more efficient by using convolutions in log-space. However, the main feature of both the FP-nets and the log-nets is that the networks learn appropriate filter pairs to be AND-combined.

FP-nets and the here proposed Min-Nets are inspired by what is known about biological vision since the seminal work of Hubel and Wiesel (Hubel & Wiesel (1965)). They discovered oriented neurons in primary visual cortex V1 but also end-stopped cells in V1 and V2. End-stopped cells extract features such as corners and line ends, provide an efficient representation of the visual input, and can be modelled as AND combinations of linear, oriented filters (Zetzsche & Barth (1990)). Motion

3rd Workshop on Shared Visual Representations in Human and Machine Intelligence (SVRHM 2021) of the Neural Information Processing Systems (NeurIPS) conference, Virtual.

selective neurons have also been modelled by using AND combinations of linear spatio-temporal filters based on the Riemann tensor (Barth & Watson (2000)).

The question of why end-stopped neurons and AND combinations of linear filters are useful for vision can be answered based on the statistics of natural images. A first useful bias is due to the fact that 0D-features, i.e., local image patches of uniform intensity or color, are both redundant and frequent in natural images. Therefore, a representation of images in terms of edges is efficient; it reduces the entropy of the representation. 1D-features such as straight edges and lines, however, are also redundant and more frequent in natural images than 2D-features, such as curved edges and corners. Therefore, a representation provided by end-stopped units is more efficient than one provided by oriented filters. Details on the above-summarized statistics and on why AND combinations are needed to exploit them can be found in Zetzsche et al. (1993).

However, we do not assume that real neurons would compute ideal multiplications and strict minimum operations, but that they may approximate such operations in some way and thus generate AND combinations of filters instead of just using point-wise non-linearities (see also Paiton et al. (2020) regarding the limits of point-wise non-linearities). Here, we present the novel Min-Net architecture that implements the minimum operation as the simplest kind of AND operation on pairs of learned filters. Moreover, minimum operations are faster than multiplications and make training easier due to more conservative gradients.

## 2   Methods

Min-Nets are CNNs containing a certain number of Min-blocks that process an input Tensor $\mathbf{T}_0$ and, via a sequence of different layers, transform them to a tensor $\mathbf{T}_{out}$. The block structure resembles the inverted residual block of the well-known MobileNet-V2 (Sandler et al. (2018)) architecture, with the important extension that it contains two filter operations that are combined by a minimum operation. In short, the block consists of (i) a convolution layer with kernel size one, batch normalization, and ReLU, increasing the number of input feature maps, (ii) two depth-wise separable (DWS) convolutions, each learning one filter per feature map with instance normalization and ReLU, the outputs of which are element-wise combined by the minimum function, (iii) a second $1 \times 1$-convolution with batch normalization and ReLU, recombining the outputs of (ii) to the desired number of feature maps and, finally, (iv) a residual connection. The next subsection gives a more detailed description. The middle panel of Figure 1 depicts a Min-block.

### 2.1   Min-blocks

First, $qd_{out}$ feature maps are computed as weighted sums of the $d$ feature maps of $\mathbf{T}_0 \in \mathbb{R}^{h \times w \times d}$. $q$ is an expansion factor, and $d_{out}$ the desired number of output feature maps. Thus, a convolution layer with kernel size one, and subsequent ReLU, computes the $(i, j)$-th pixel of the $m$-th feature map as:

$$\mathbf{T}_1[i, j, m] = ReLU(\sum_{n=1}^{d_{in}} w_m^n \mathbf{T}_0[i, j, n]); m = 1, ..., qd_{out}. \tag{1}$$

Note that before the ReLU, $\mathbf{T}_1$ is normalized via batch normalization. Next, two DWS convolutions are applied: for each feature map $\mathbf{T}_1^m$ a filter-pair $\mathbf{V}_m$ and $\mathbf{G}_m \in \mathbb{R}^{k \times k}$ is learned. For each pixel $(i, j)$, the filter operation computes the scalar product of the filter vectors (i.e., the vectorized filter kernels $vect(\mathbf{V}_m) = \mathbf{v}_m$ and $\mathbf{g}_m \in \mathbb{R}^{k^2}$) and the vectorized feature map patch $\mathbf{x}_m \in \mathbb{R}^{k^2}$ with $(i, j)$ being the pixel-coordinates of the center pixel. Subsequently, instance normalization (Ulyanov et al. (2016)) is applied with $\mu_{v_m}$ and $\sigma_{v_m}$ being the mean and standard deviation of the feature map $\mathbf{T}_1^m$ filtered with $\mathbf{v}_m$. Again, a ReLU is applied afterwards. Using the min-function, we combine these two intermediate outputs to $\mathbf{T}_2 \in \mathbb{R}^{\frac{h}{s} \times \frac{w}{s} \times qd_{out}}$, $s$ being the stride of the DWS-convolution that allows for subsampling. The Min-unit, as the main building block of the proposed network, is defined as:

$$\mathbf{T}_2[i, j, m] = \min \left[ \frac{ReLU(\mathbf{x_m}^T \mathbf{v_m} - \mu_{v_m})}{\sigma_{v_m}}, \frac{ReLU(\mathbf{g_m}^T \mathbf{x_m} - \mu_{g_m})}{\sigma_{g_m}} \right]. \tag{2}$$

Note that such a Min-unit, or Min-neuron, will have a significant output only if both filters $\mathbf{v_m}$ and $\mathbf{g_m}$ are activated.

A second linear recombination with batch normalization and ReLU (see Equation 1) creates the tensor $\mathbf{T}_3 \in \mathbb{R}^{\frac{h}{s} \times \frac{w}{s} \times d_{out}}$. Finally, a residual connection adds the original input to $\mathbf{T}_3$:

$$\mathbf{T}_{out} = \mathbf{T}_0 + \mathbf{T}_3. \tag{3}$$

Some additional computations need to be applied if the dimensions of $\mathbf{T}_0$ and $\mathbf{T}_3$ do not match. See the Appendix for details.

## 2.2 Transforming state-of-the-art CNNs to Min-Nets

In our experiments, we evaluate how the addition of Min-blocks changes the performance of state-of-the-art CNNs. Networks such as the DenseNet (Huang et al. (2017)) and the ResNet (He et al. (2016)) have a similar overall structure: after an initial convolution layer, the networks consists of several *stacks* that are larger structures containing a number of *blocks*. Analogously, a block contains a number of *layers*, for example, convolutions, batch normalization, and non-linearities. After the last stack, global average pooling is applied with a subsequent linear layer and softmax to compute the class probabilities. Stacks are separated by a downsampling layer (strided convolution or a max-pooling layer), except for the first stack after the initial convolution. When applied to smaller datasets, e.g., Cifar, both the DenseNet and ResNet consist of three stacks, containing $N$ blocks each, with the input heights and widths $32 \times 32$, $16 \times 16$, and $8 \times 8$. Thus, only the second stack and third stack employ downsampling. We use a simple design rule to enrich these architectures with blocks that directly compute pairwise AND-combinations: *Each first block of a stack is substituted by a Min-block.* Note that this approach was derived empirically. In preceding experiments, other combinations of Min-blocks and convolutional blocks were tested but yielded somewhat inferior or equal results. We chose a ResNet implementation that uses pyramid blocks (Han et al. (2017)) since it performs better than the original ResNet and is currently often used for different applications. Here, the sequence of layers consists of batch normalization, $3 \times 3$ convolution, batch normalization, ReLU, $3 \times 3$ convolution, and batch normalization. The sequence output is added to the original input of the block. As a second reference network, we use the DenseNet-BC. In this architecture, each so-called bottleneck block consists of batch normalization, ReLU, and a $1 \times 1$ convolution layer, followed by a second batch normalization and ReLU and a $3 \times 3$ convolution layer. A comparison of the different block structures is given in Figure 1.

## 2.3 Hyperselectivity

CNNs employ linear neurons with points-wise non-linearities (LN-neurons), e.g., a ReLU that is applied to each pixel independently. Thus, the direction of maximal activation, i.e., the optimal stimulus, $\mathbf{x}^*$ points in the same direction as the neuron's weight vector $\mathbf{w}$. Any deviation $\mathbf{o}$ orthogonal to the optimal stimulus does not alter the neuron's output: $ReLU(\mathbf{w}^T(\mathbf{x}^* + \mathbf{o})) = ReLU(\mathbf{w}^T \mathbf{x}^*)$. In contrast, a neuron $f(\mathbf{x})$ is hyperselective, if there exist orthogonal perturbations, that reduce the output:

$$\text{f is hyperselective} \iff \exists \mathbf{o} : f(\mathbf{x}^* + \mathbf{o}) < f(\mathbf{x}^*); \mathbf{x}^{*T}\mathbf{o} = 0 \tag{4}$$

For a Min-neuron (see Equation 2), the optimal stimulus is the bisector of the angle $\gamma$ between the filter vectors $\mathbf{v}$ and $\mathbf{g}$. Any perturbation within the plane spanned by $\mathbf{v}$ and $\mathbf{g}$ that is orthogonal to the optimal stimulus reduces the neuron's output, because one of the scalar products is smaller than it would be without the perturbation, e.g., $\mathbf{v}^T(\mathbf{x}^* + \mathbf{o}) < \mathbf{v}^T \mathbf{x}^*$. Hyperselective neurons are harder to activate by signals that differ from the optimal stimulus. Thus, unwanted perturbations, such as those due to adversarial examples (Szegedy et al. (2013); Goodfellow et al. (2014)) or compression artifacts, have a smaller impact on the neuron's – and consequently, the whole network's – output. Note that an LN-neuron located in the second or higher layer of a CNN can become hyperselective indirectly when regarding the pixel space as input. In contrast, a Min-neuron is directly hyperselective to its input.

## 3 Experiments

We hypothesize that Min-Nets introduce a useful inductive bias to a CNN architecture: the minimum of a learned filter-pair, a Min-neuron, explicitly models end-stopped neurons and leads to efficient

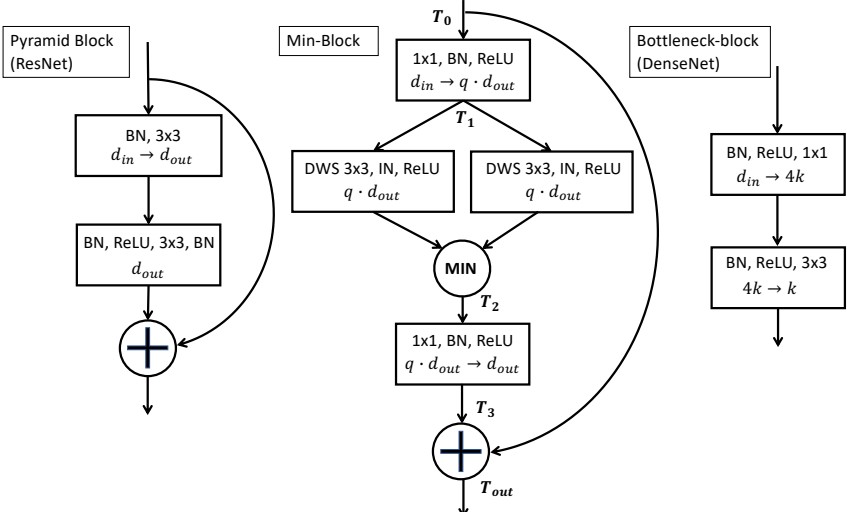

Figure 1: Comparison of different block architectures. The left scheme shows the basic block of a ResNet adopted from the PyramidNet. The middle scheme shows the proposed Min-block, the right scheme the DenseNet's bottleneck block. Arrows show the data flow; the abbreviations within the boxes denote different layers and operations. BN: batch normalization, IN: instance normalization, $3 \times 3$: a convolution with kernel size 3, DWS $3 \times 3$: a depthwise-separable convolution with kernel size 3. $d_{in} \to d_{out}$ expresses a change in feature maps from $d_{in}$ to $d_{out}$. $k$ is the growth factor, $q$ is the expansion factor. The plus in the circle is a residual connection, the min in the circle a minimum operation of two inputs. $\mathbf{T}_i$ denotes the (intermediate) tensor results (see for example Equation 2). Note that instead of using residual connections, the DenseNet concatenates feature maps from previous layers – which is not depicted here.

representations, which are adapted to the statistics of natural images, thus leading to better generalization. Furthermore, unlike traditional model neurons, Min-neurons are hyperselective, a property that should make them more robust, e.g., to compression artifacts. We ran two experiments to investigate these conjectures.

## 3.1 Classification performance on Cifar-10

We compared models with different depths created from two reference network architectures: the ResNet and the DenseNet. For each model, we created a Min-Net version using our design rule. We compared models with different depths, defined by the number of blocks per stack. For the ResNets, we created models with $N \in \{3, 5, 7, 9\}$ blocks. We trained DenseNets with $N \in \{3, 9, 16\}$ blocks and a growth-rate $k = 12$, corresponding to networks $(L = 22, k = 12)$, $(L = 58, k = 12)$, and $(L = 100, k = 12)$ in the DenseNet paper's naming convention. For all models, we set the expansion factor $q = 2$, and we trained and tested the models on Cifar-10 (Krizhevsky et al. (2021)) with a standard procedure for data augmentation (see Appendix). Each ResNet was trained for 200 epochs with a learning rate of 0.1, a weight decay of 0.0001, and using SGD with a momentum of 0.9. The batch size was 128. After 100 and 150 epochs, the learning rate was divided by 10. Each net was trained for five random seeds, altering the batch order and random initialization. We trained the DenseNets for 300 epochs, where the initial learning rate of 0.1 was divided by ten after 150 and 225 epochs. Instead of 128, a batch size of 64 was used. In addition, we used the Nesterov momentum. We trained each network with three different random seeds. All other parameters were equal to the ResNet experiments. Note that we did not employ an early stopping strategy but rather report the final test results after 200 or 300 epochs, respectively.

## 3.2 Robustness to compression artifacts

Since Min-Nets have neurons that are hyperselective, one can expect the networks to be more robust against perturbations. To evaluate robustness, in a second experiment, each model trained in the first experiment was tested on JPEG-compressed versions of the Cifar-10 test set $\mathcal{X}_{test}$. We created

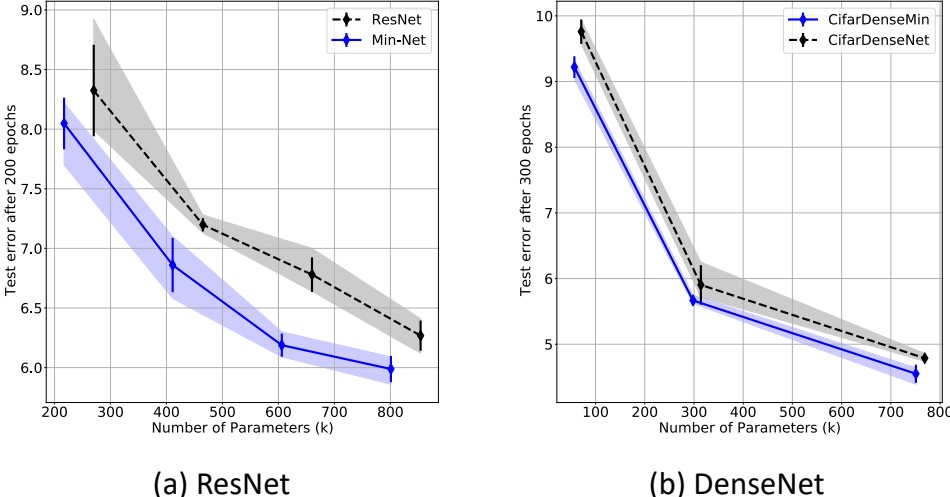

(a) ResNet        (b) DenseNet

Figure 2: Cifar-10 test error results. Black dashed lines show the baseline results – solid blue lines the Min-Net results. Each diamond denotes the mean error of a model with a specific number of blocks. The error bars indicate the standard deviation of each model. The colored areas show the distance between the minimum- and maximum values. The x-axis shows the number of parameters, increasing with the number of blocks. The y-axis is the test error after 200 and 300 epochs for the ResNets (a) and DenseNets (b), respectively.

nine additional test sets $\mathcal{S}_Q = \{JPEG(\mathbf{I}, Q) : \mathbf{I} \in \mathcal{X}_{test}\}$ with quality $Q \in \{90, 80, ..., 10\}$, $JPEG(\mathbf{I}, Q)$ being the compression function for an input image $\mathbf{I}$. For $Q = 100$, the original image is returned (no compression); $Q = 1$ would yield maximal compression with images that are barely recognizable (see Figure 5 in the Appendix for examples). As robustness metric, we report the percentage of changed predictions (POCP), i.e., we compared the model's predictions with full quality to the predictions on $\mathcal{S}_Q$ and computed the number of changed predictions divided by the number of images (see Eq. 5 in the Appendix). Using this metric, we avoid offsets introduced by models with lower test errors. We used the JPEG algorithm from the OpenCV (Bradski (2000)) library.

## 4 Results

The Cifar-10 test errors of each model are shown in Figure 2. Note that in all configurations, the Min-Net has a lower mean test error than the corresponding baseline networks while using fewer parameters.

Next, we analyse the robustness of the networks. Figure 3 shows the POCP (Equation 5) for the quality levels $Q = 90, 80, 70, 60$. Compared to the ResNet and DenseNet, the Min-Nets are less likely to change their output when presented with compressed images. However, even the more robust Min-Nets remain sensitive to compression artifacts given that $8\%$ of the predictions change with a slightly altered test set (such as $\mathcal{S}_{90}$).

## 5 Discussion

We have presented a novel network architecture and have demonstrated its competitive performance. We have designed experiments with state-of-the-art and popular deep networks and showed that we could improve their performance by substituting original blocks in the network architecture with Min-blocks that implement a minimum operation on pairs of feature maps. Note that by using a simple design rule, any traditional network can easily be transformed into a Min-Net that will most likely perform better. Also, note that we did not tune hyperparameters specific to the Min-Nets but used the hyperparameters of the original networks; additional tuning may thus lead to even better performance of the Min-Nets.

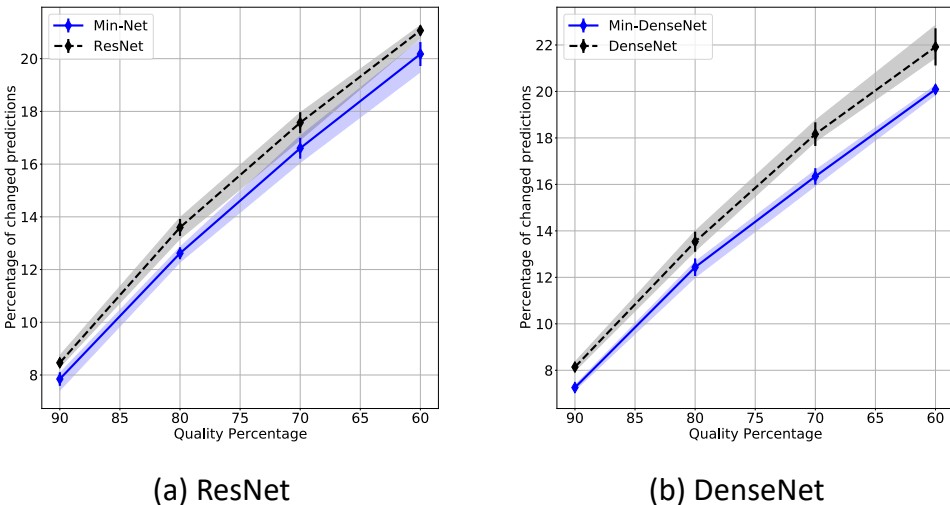

|     |     |
| --- | --- |
| (a) ResNet | (b) DenseNet |

Figure 3: POCP depending on JPEG quality: the x-axis shows the quality parameter $Q$ for the JPEG compression algorithm (which can vary from 100 to 1). The y-axis denotes, in relative terms, how many predictions were different from the original predictions (with $Q = 100$) when using a more compressed test set.

We believe that a bias that is appropriate for natural images leads to the improvements that we have obtained. As argued in the Introduction, this bias allows the network to learn more efficient representations based on model neurons that are end-stopped.

The pairwise minimum operations that we introduce allow for AND rather than OR combinations of features and make the resulting neurons more selective than linear filters with only point-wise non-linearities. The demonstrated increased robustness of the Min-Nets is most likely due to the fact that Min-units are hyperselective. It has been argued before that hyperselectivity might be the key to greater robustness (Paiton et al. (2020)).

In previous work, similar results have been obtained with Log-nets and FP-nets where logarithms and feature products have been used instead of the minimum. This shows that the key to the improvements demonstrated here is mainly the AND combination of filters pairs and less the way in which the AND is implemented. Nevertheless, we are looking forward to more general and even more effective network architectures that may emerge based on the principle of using the minimum activation in different feature maps.

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

# A   Appendix

## A.1   Residual connections

If a stride of two is used ($s = 2$), average pooling with a kernel size of two and stride of two is applied to $\mathbf{T}_0$. If $d$ is smaller than $d_{out}$, $\mathbf{T}_0$ is padded with ($d_{out} - d$) all-zero feature maps. If $d$ is larger than $d_{out}$, a linear recombination without ReLU and batch normalization computes a reduced version of $\mathbf{T}_0$ with $d_{out}$ feature maps.

## A.2   Data augmentation

During training, the images were flipped horizontally with a 50% chance. Furthermore, they were padded with four pixels on each side and randomly cropped using a $32 \times 32$ window. We normalized each channel of an input image by subtracting the channel's mean pixel and dividing by the standard deviation, all derived from the training set.

## A.3   Percentage of changed predictions

We computed the POCP depending on $Q$ like this:

$$\text{POCP}(f, Q) = \frac{1}{|\mathcal{X}_{C10}|} \sum_{\mathbf{I} \in \mathcal{X}_{C10}} \mathbb{1}(f(\mathbf{I}) \neq f(JPEG(\mathbf{I}, Q))); \tag{5}$$

$\mathbb{1}(S)$ returns a 1 if $S$ is true, as 0 otherwise; $f(\mathbf{I})$ is the model's class prediction for the input image $\mathbf{I}$.

## A.4   Technical details for training

All experiments were conducted with Pytorch (BSD license) (Paszke et al. (2019)) version "1.9.0+cu102", on a standard computer running on Linux using Nvidia RTX 2080 GPUs with CUDA version 11.2. Training a DenseNet ($L = 100, k = 12$) took roughly 21 hours, training a ResNet ($N = 9$) 3 hours. The code creating the ResNet was adapted from Idelbayev (2020) (BSD 2-Clause "Simplified" License) and from Han et al. (2017)[1]. We adapted the DenseNet code[2] from Pleiss et al. (2017) (MIT License). OpenCV uses the Apache 2 license. We published our code on Github[3]

## A.5   Additional results: Densenet

Table 1 shows additional results for the DenseNet models on the Cifar-10 classification task. When regarding minimal errors over all epochs, i.e., the best result with early stopping, we obtained similar results for the DenseNet ($L = 100, k = 12$) as reported in the original paper (4.51%): the minimum error reached by the baseline DenseNet was 4.56% and 4.36% for the Min-Net.

## A.6   Additional results: JPEG compression

Figure 4 shows the Cifar-10 test errors depending on the $Q$ value. Note that these plots only provide the results for $Q = 100, 90, ..., 60$, and only for the largest models of each experiment (ResNet: $N = 9$, DenseNet: $k = 12, L = 100$). The results for $Q = 100, 90, ..., 10$ and for all models are given in the Tables 2,3, 4, and 5. Table 2 shows the test error values for the DenseNet experiment and Table 3 shows the POCP. The test error and POCP for the ResNet experiments are given in the Tables 4 and 5, respectively. An example image compressed with different $Q$-values is shown in Figure 5. Note that the POCP of a specific $Q^*$ can be higher than the respective increase of the test error. For example, $\Delta(Q^*)$ – i.e., the test error for $Q^* = 90$ minus the original error – is lower than the POCP for $Q = 90$ since the POCP measures any change in classification: apart from correct classifications for $Q = 100$ being incorrect for $Q^*$, the POCP also counts classifications that are already incorrect and are now classified as another false class. Furthermore, a few previously incorrect examples can become correct and decrease $\Delta(Q)$.

---

[1]`https://github.com/dyhan0920/PyramidNet-PyTorch`

[2]`https://github.com/gpleiss/efficient_densenet_pytorch`

[3]`https://github.com/pgruening/bio_inspired_min_nets_improve_the_performance_and_robustness_of_deep_networks`

| Model | k | # Parameters | L | After 300 epochs | | | | Min. over all epochs | | | |
|---|---|---|---|---|---|---|---|---|---|---|---|
| | | | | mean | std | min | max | mean | std | min | max |
| Min-Net | 12 | 751786 | 100 | 4.55 | 0.14 | 4.39 | 4.64 | 4.51 | 0.13 | 4.36 | 4.59 |
| DenseNet | 12 | 769162 | 100 | 4.79 | 0.06 | 4.74 | 4.86 | 4.62 | 0.06 | 4.56 | 4.66 |
| Min-Net | 12 | 298564 | 58 | 5.67 | 0.07 | 5.62 | 5.75 | 5.51 | 0.07 | 5.46 | 5.59 |
| DenseNet | 12 | 314470 | 58 | 5.90 | 0.30 | 5.72 | 6.25 | 5.65 | 0.14 | 5.51 | 5.78 |
| Min-Net | 12 | 57040 | 22 | 9.22 | 0.17 | 9.03 | 9.33 | 8.78 | 0.10 | 8.72 | 8.89 |
| DenseNet | 12 | 71686 | 22 | 9.76 | 0.19 | 9.57 | 9.94 | 9.39 | 0.13 | 9.27 | 9.52 |

Table 1: Classification results for the DenseNet experiment.

| Model | L | k | Q=100% | 90% | 80% | 70% | 60% | 50% | 40% | 30% | 20% | 10% |
|---|---|---|---|---|---|---|---|---|---|---|---|---|
| DenseNet | 22 | 12 | 9.8 | 15.4 | 20.0 | 24.0 | 27.6 | 30.8 | 34.6 | 39.5 | 47.8 | 64.6 |
| DenseNet | 58 | 12 | 5.9 | 10.9 | 15.7 | 20.0 | 24.3 | 27.3 | 31.3 | 36.4 | 45.9 | 63.0 |
| DenseNet | 100 | 12 | 4.8 | 9.9 | 14.6 | 19.1 | 22.8 | 25.9 | 30.1 | 35.2 | 44.5 | 62.6 |
| Min-Net | 22 | 12 | 9.2 | 14.9 | 19.5 | 23.2 | 26.4 | 29.5 | 32.5 | 37.5 | 44.9 | 60.3 |
| Min-Net | 58 | 12 | 5.7 | 10.4 | 15.0 | 19.0 | 22.6 | 25.4 | 28.6 | 34.0 | 43.1 | 62.5 |
| Min-Net | 100 | 12 | 4.6 | 8.8 | 13.6 | 17.2 | 20.9 | 23.8 | 27.6 | 33.1 | 42.5 | 61.4 |

Table 2: DenseNet: Mean test error on Cifar-10 for JPEG-compressed test sets $\mathcal{S}_Q$

.

| Model | L | k | Q=90% | 80% | 70% | 60% | 50% | 40% | 30% | 20% | 10% |
|---|---|---|---|---|---|---|---|---|---|---|---|
| DenseNet | 22 | 12 | 12.1 | 17.6 | 21.9 | 25.8 | 29.3 | 33.4 | 38.3 | 46.8 | 64.2 |
| DenseNet | 58 | 12 | 9.2 | 14.7 | 19.3 | 23.5 | 26.4 | 30.5 | 35.7 | 45.3 | 62.7 |
| DenseNet | 100 | 12 | 8.1 | 13.5 | 18.2 | 21.9 | 25.1 | 29.3 | 34.8 | 44.1 | 62.6 |
| Min-DenseNet | 22 | 12 | 11.7 | 16.9 | 21.4 | 24.8 | 28.0 | 31.2 | 36.3 | 43.9 | 60.0 |
| Min-DenseNet | 58 | 12 | 8.3 | 13.7 | 17.8 | 21.6 | 24.6 | 27.9 | 33.5 | 42.5 | 62.4 |
| Min-DenseNet | 100 | 12 | 7.3 | 12.4 | 16.3 | 20.1 | 23.2 | 27.1 | 32.6 | 42.1 | 61.4 |

Table 3: DenseNet: Mean number of changed predictions on Cifar-10 for JPEG-compressed test sets $\mathcal{S}_Q$

.

| Model | N | 100% | 90% | 80% | 70% | 60% | 50% | 40% | 30% | 20% | 10% |
|---|---|---|---|---|---|---|---|---|---|---|---|---|
| Min-Net | 3 | 8.0 | 12.5 | 16.7 | 20.2 | 23.5 | 26.2 | 29.3 | 34.1 | 42.5 | 59.1 |
| Min-Net | 5 | 6.9 | 11.3 | 15.1 | 18.7 | 22.0 | 24.9 | 27.9 | 33.4 | 42.3 | 59.2 |
| Min-Net | 7 | 6.2 | 10.6 | 14.6 | 17.9 | 21.5 | 24.5 | 27.7 | 32.8 | 42.2 | 59.9 |
| Min-Net | 9 | 6.0 | 10.1 | 14.4 | 18.0 | 21.4 | 24.2 | 27.4 | 32.7 | 41.7 | 58.8 |
| ResNet | 3 | 8.3 | 13.2 | 17.4 | 21.0 | 24.5 | 27.6 | 31.1 | 36.0 | 44.1 | 59.8 |
| ResNet | 5 | 7.2 | 11.8 | 16.3 | 20.0 | 23.6 | 26.6 | 30.0 | 35.4 | 44.0 | 59.6 |
| ResNet | 7 | 6.8 | 11.4 | 15.9 | 19.2 | 22.9 | 26.3 | 29.7 | 35.0 | 43.7 | 60.1 |
| ResNet | 9 | 6.3 | 10.9 | 15.3 | 19.0 | 22.4 | 25.6 | 29.2 | 34.5 | 43.4 | 59.1 |

Table 4: ResNet: Mean test error on Cifar-10 for JPEG-compressed test sets $\mathcal{S}_Q$

.

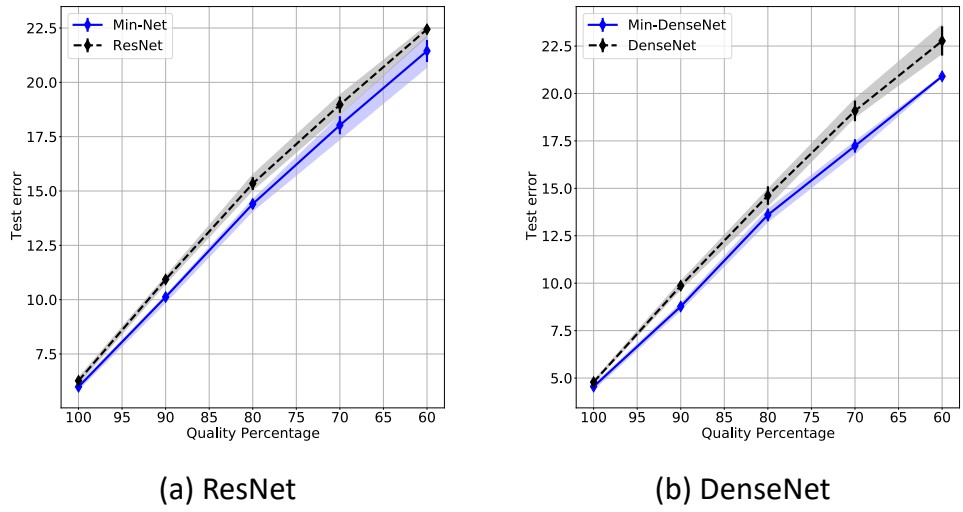

| (a) ResNet | (b) DenseNet |

Figure 4: Cifar-10 test error depending on the compression quality.

| Model | N | 90% | 80% | 70% | 60% | 50% | 40% | 30% | 20% | 10% |
|---|---|---|---|---|---|---|---|---|---|---|
| Min-Net | 3 | 9.5 | 14.5 | 18.4 | 21.8 | 24.7 | 28.0 | 33.1 | 41.8 | 58.8 |
| Min-Net | 5 | 8.5 | 13.2 | 17.1 | 20.6 | 23.7 | 26.8 | 32.5 | 41.6 | 58.8 |
| Min-Net | 7 | 8.1 | 12.8 | 16.7 | 20.3 | 23.3 | 26.6 | 32.0 | 41.4 | 59.7 |
| Min-Net | 9 | 7.8 | 12.6 | 16.6 | 20.2 | 23.0 | 26.3 | 31.7 | 40.9 | 58.5 |
| ResNet | 3 | 10.1 | 15.1 | 19.3 | 22.9 | 26.2 | 29.9 | 34.8 | 43.3 | 59.3 |
| ResNet | 5 | 9.4 | 14.4 | 18.6 | 22.3 | 25.5 | 29.1 | 34.5 | 43.2 | 59.3 |
| ResNet | 7 | 8.9 | 14.0 | 17.9 | 21.8 | 25.1 | 28.8 | 34.2 | 43.1 | 59.8 |
| ResNet | 9 | 8.5 | 13.6 | 17.6 | 21.1 | 24.5 | 28.2 | 33.6 | 42.7 | 58.7 |

Table 5: ResNet: Mean number of changed predictions on Cifar-10 for JPEG-compressed test sets $\mathcal{S}_Q$

.

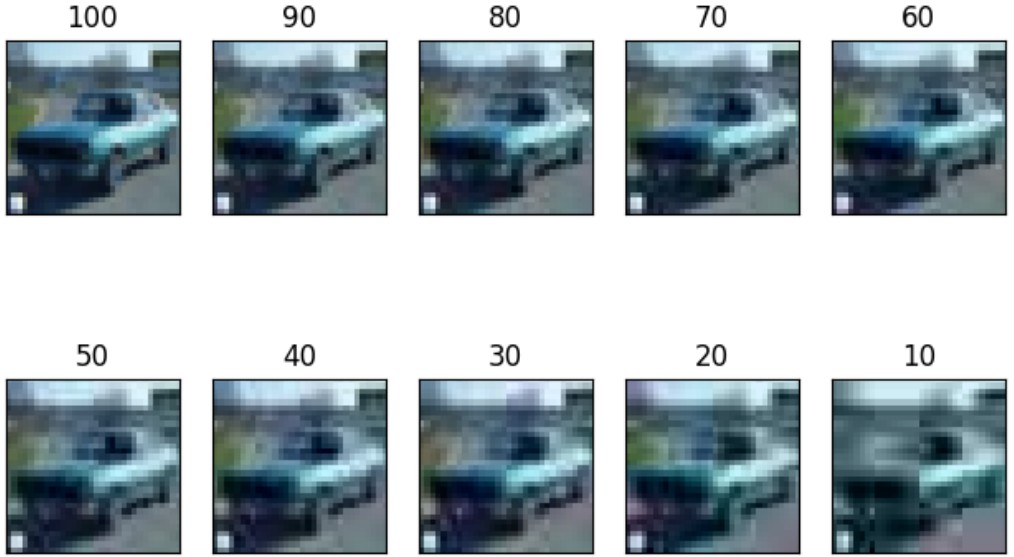

Figure 5: JPEG-compression example for different $Q$ values.

