# OpenReview forum: "Bio-inspired Min-Nets Improve the Performance and Robustness of Deep Networks"
_NeurIPS.cc/2021/Workshop/SVRHM — SVRHM 2021 Poster_

### Official Review · Reviewer_7MAZ · 2021-10-14
**Neurally inspired modification to existing CNNs leads to improved results**

**Rating:** 7
**Confidence:** 3

**Review:**

I enjoyed reading this paper and think that exploring the addition of new, neurally inspired DNN architectural choices is an interesting and worthy goal. This submission also certainly fits into the scope of the workshop, as insight from visual neuroscience was a clear motivator. For these reasons, I recommend the submission to be accepted as is. However, I think addressing point 3 below will make it a stronger, more impactful paper.

Strengths:
1. The rationale for adding end-stop neuron like units into DNNs used in computer vision was motivated;
2. The implementation of this new method was detailed (both in the main text and the appendices);
3. The results were interesting and encouraging. Additionally, it was clear that Min-blocks could be extended to a variety of architectures, beyond what was tested;
4. There is clear potential for future directions.

Weaknesses:
1. The fact that Min-Nets exhibited similar scaling in POCP as ResNets and DenseNets (despite slightly better robustness) slightly dampens the results;
2. The fact that Log-nets and FP-nets have had similar benefits using similar mechanisms [lines 180-181], also slightly dampens the novelty of this work;
3. There are two points that are brought up in the submission that are particularly interesting, but not fully addressed. I understand that the page limit constrains discussion, but maybe it could be included in the appendices? These two points are: 1) why do Min-Nets exhibit similar scaling in POCP as ResNets and DenseNets? and 2) why does substituting the first block with a Min-block make more of a difference than substituting a later block? I don’t think a full answer is necessary, but possible hypotheses (especially those that draw influence from visual neuroscience) would be interesting and insightful.

One final minor point. I am not familiar with FP-nets, and I imagine others might not be as well. It would be helpful if you would define what the acronym is the first time you use it [line 22].

---

> ### Author Response · Authors · 2021-12-03
> **Response to reviewer 3**
>
> We would like to elaborate on the two major points that the reviewer makes.
>
> (1) The reviewer notes that in Figure 2 the MinNet- and baseline POCP-curves increase similarly with decreasing input image quality. We think that, to some degree, this impression is due to the large scale of the plots. Considering the error-plots in the appendix, without artifacts, the improvement is around 0.3% and it is then increased up to about 2% on the compressed dataset, which is quite significant.
> Moreover, an increase in classification errors for any network was predictable, as some of the strongly compressed Cifar-10 become very hard to classify – even for a human. Still, a MinNet is far from the levels of human performance regarding robustness.
> One reason may be that only a few of the neurons of an entire network exhibit hyper-selective properties. For example, for the largest ResNet, each stack contains one Min-block relative to eight baseline blocks. However, we still need to quantify further how hyper-selectivity increases robustness. In addition, if many hyper-selective neurons would indeed be beneficial, there is still the issue of finding a network that contains more Min-blocks while maintaining or improving the performance of the baseline.
>
> (2) The reviewer is missing an explanation of why the placing of Min-blocks at specific positions makes a significant difference. Indeed, the question of where to place a Min-block needs further investigation but comprehensive experiments are difficult to implement because the number of possibilities can easily reach 2 to the power of 20 or more. For a subset of these possible configurations, the differences in performance are rather small, i.e., the networks are only somewhat sensitive regarding the positioning of the blocks. However, what we know is that (i) using fewer Min-blocks than conventional blocks and (ii) substituting blocks in early positions in the network is beneficial (in accordance with the fact that end-stopped neurons are found in rather early layers such as V2).

---

### Official Review · Reviewer_kAaQ · 2021-10-27
**Good article, more qualitative analysis would have been interesting**

**Rating:** 8
**Confidence:** 3

**Review:**

### Summary :
The authors propose to include a new non-linear operation in the state of the art deep learning architecture. This operation is implemented with the minimum operation between 2 filters (the effect is similar to an AND operation). This operation leads to ‘end-stopped’ neurons (neurons encoding for a corner …) that have been found in the early visual cortex. In terms of performance, this min operation allows a better robustness to the jpeg compression artifact and improve the testing classification accuracy significantly.


### General comments :
This article is in general well written and pleasant to read. The authors' claims in terms of improved classification accuracy and better robustness are clear, unambiguous and  backed by relevant experiments. I would have appreciated the section on the min-block (i.e. section 2.1) to be clearer (and ideally illustrated with a comparative scheme with the ResNet block). Given the short format of this workshop paper, and the promising results I would tend to accept the article. Please find more comments below (mostly suggestion to improve the current version of the article). My rating : 8

### More specific comments
* It is a bit disappointing that the section that is crucial for the understanding of the paper is the less clear one. I would suggest to include a scheme that is comparing, side by side, a ResNet block with a Min-Block. I would also suggest to rewrite the section 2.1. such that it is highlighting the main difference with ResNet blocks.

* The authors have successfully demonstrated that the MinBlock is bringing robustness to jpeg compression artifact. But is it also showing an increased robustness to different types of perturbation ? It would be interesting to run a systematic analysis of the network robustness with the classical adversarial attacks.

### Suggestion of improvement for future work
* The authors say that the AND operation leads to filters similar to those encountered in the early visual cortex (e.g. end-stopped cells). It would be interesting to show it qualitatively. For example, one can train a Min-network and a standard one on natural images dataset and then visualize the learned receptive fields (i.e. the projection in the input space of the filters located at different depth). Then, it would interesting to run a qualitative analysis of the differences between the 2 set of filters.

* Another Interesting way of illustrating the end-stopped neurons, would be to present a stimulus composed of numerous corner, line-ends... (at test time, the training would be standard) and to see if more neurons are responding to this kind of stimuli in the Min-network compared to a classical one.

* The shown results (both improved classification accuracy and better robustness) should be generalized on more network architectures and more importantly on different datasets (e.g. higher resolution image dataset).

* Another interesting analysis would be to analysis the sample image misclassified by the classical network and well-classified by the Min-Network. Is there any more end-stops patterns in those samples ?

---

> ### Author Response · Authors · 2021-12-03
> **Response to reviewer 2**
>
> As requested, we have now included a comparative scheme of the three different blocks.
> We agree with the reviewer that more qualitative analysis would have been interesting but we did not have the space for it. However, we provided such analysis in the above-mentioned paper in the Journal of Vision. For example, we showed that FP-nets based on different ResNets were more robust against the well-known fast gradient sign method (FGSM). Furthermore, we could show that FP-neurons are more likely to be selective to 2D features, such as corners, compared to standard CNN neurons. Furthermore, we showed that hyper-selectivity directly depends on the angle between the two filters v and g, and we could demonstrate that the entropy of a feature map is reduced with increasing angle. FP-nets differ from Min-Nets in that the AND is implemented by multiplication instead of minimum operation. We have obtained similar results with Min-nets but did not have the space to publish them here.
> However, an in-depth analysis of the different misclassified images and of the receptive fields remains for future research.

---

### Official Review · Reviewer_c7ES · 2021-10-30
**Promising results with questionable interpretation and biological relevance**

**Rating:** 7
**Confidence:** 3

**Review:**

The authors tested a simple modification of popular CNN architectures (ResNet and DenseNet) that led to moderate improvements on image classification (CIFAR-10) and one measure of robustness (robustness to JPEG compression artefacts). The authors consider the key of their modification to be a minimum operation between pairs of feature channels within one layer (see below for alternative explanations) and claim that this minimum operation relates to end-stopping in biological neurons.

I find the mild improvement on two measures (classification and robustness to JPEG compression) somewhat interesting, given the simplicity of the architectural change. It is laudable that the authors controlled for number of model parameters while comparing performance. The paper states that Min-nets may be a computationally efficient alternative to Log-nets and FP-nets, although the paper does not make direct comparisons with these alternatives. These contributions notwithstanding, I do not think the results are necessarily due to the minimum operation, as the authors would like to claim. I also find the liberal analogies to biology (and other work in computer vision) insufficiently explained and possibly inaccurate.

My foremost concern is that improved performance of Min-nets may be due to increased depth, both directly and indirectly. Based on the text, each Min-block comprises 3 convolutional layers (Secs. 2 and 2.1), whereas the corresponding block it replaces in the control ResNet or DenseNet comprises only 2 convolutional layers. Moreover, the minimum operation is itself a nonlinear operation, and thus increases the effective nonlinearity, or depth, of the network. It is not clear whether the increased performance can be attributed to the direct increase in depth---thus bearing no relation to the minimum operation---or to the indirect increase in depth, which does arise from the minimum operation, but not necessarily in a biologically relevant way.

This connects to the second concern, that the biological relevance (and other analogies) claimed by the paper may be inaccurate and misleading. The paper repeatedly suggests that end-stopping is analogous to the AND (and thus MIN) operation, and even to curvature/corner tuning (lines 41--44). End-stopping describes the property that responses of, say, an orientation-selective cell to a straight bar, increases only up to a certain bar length, then decreases if the bar is further lengthened. (Thus, the maximum response is elicited when the bar is 'end-stopped.') This property does not necessarily imply an AND operation, or curvature/corner tuning: A bar fading out at the edge or terminated with any border (other than a straight edge with two right-angle corners) should also strongly activate an end-stopped cell, so no particular feature need to be AND-combined with the bar's orientation. Although end-stopping *can* be modeled by an AND operation, and although the MIN operation represents *some* AND operation, it is far from certain that a MIN operation with learned weights should represent end-stopping at all. End-stopping can be measured, but the paper has not done so.

The paper also makes an appeal to hyperselectivity as defined in Paiton et al., 2020 (which follows Vilankar & Field, 2017, the latter itself a meeting abstract that contains few details). First of all, hyperselectivity in Paiton et al. is defined in pixel space, by which standard any unit in a CNN past the first layer can potentially be hyperselective. The minimum operation guarantees hyperselectivity, but only along one dimension---all dimensions orthogonal to the plane defined by **v**, **g** (the pair of filters being MIN-combined) are not affected. This is far from the case in Paiton et al., where hyperselectivity is induced by local competition among all neurons in a layer, and thus affects many more dimensions. As above, hyperselectivity can be measured, but the paper has not done so. Along the same line, the paper contains several more gratuitous analogies to Gaussian curvature, product of eigenvalues, etc. (lines 12--21), all described too vaguely to be justifiable or useful.

Despite these criticisms, given the relative simplicity of adding a min operation, I think the results, and the reason for these results, may merit discussion in the workshop.

---

> ### Author Response · Authors · 2021-12-03
> **Response to reviewer 1**
>
> One concern is that the Min-nets might perform better because they are deeper than the baselines since we use three convolution (conv-) layers in the Min-block instead of just two convolutions as in the baseline block. We agree with the reviewer that a more similar structure would lead to a better comparison between the Min-nets and the baseline networks. However, since we only alter three blocks for each model, the entire network is deepened by only three additional layers, which is just a five percent increase for a 50-layer CNN. Furthermore, we do not think that only the fact that we use three layers would improve the network's capacity over two layers, because we do not use three "full-size" conv-layers: a Min-block only uses two 1x1 conv-layers, essentially only linear combinations of the feature maps, and a depth-wise-separable (DWS) convolution, where only one filter is learned per feature map. Accordingly, a Min-block has fewer parameters and – compared to the ResNet’s block – an even smaller receptive field.
> Moreover, using [FP-blocks](https://jov.arvojournals.org/ss/deepneuralnets.aspx), we were able to improve the performance of MobileNet-V2 networks on ImageNet, and the above concerns do not apply to MobileNets.
>
> The reviewer also raises the concern that the network's effective depth would be increased by using more nonlinearities, mainly the minimum operation. In [previous work](https://arxiv.org/abs/2008.07930), we showed in an ablation study that using a single DWS conv-layer with subsequent ReLU yielded worse results than a pairwise multiplication of two DWS conv-layers. The result indicates that “simply using an additional nonlinearity” is not leading to an improvement. The AND combination of two filters seems to be essential since it makes a layer more selective. Furthermore, several small improvements of the original ResNet-blocks over the years (a good overview can be found in [Han et al.](https://arxiv.org/abs/1610.02915)), have been achieved with a reduced number of ReLUs.
>
> Considering Paiton's definition of hyper-selectivity, we have acknowledged in the camera-ready version that linear neurons from deeper layers can be hyper-selective when regarding the pixel space as input.
> The reviewer correctly observed that for a Min-neuron, hyper-selectivity only happens in the plane spanned by v and g. As this is a relatively small increase in hyper-selectivity, that is furthermore employed by only a subset of neurons in the CNN, the increase in robustness is remarkable. In the future, we will conduct further research combining more than two filters which can be done efficiently, for example, using [logarithms](https://link.springer.com/chapter/10.1007/978-3-030-61616-8_7). One would expect a further increase in robustness. However, combining more than two filters may be harder to optimize.
>
> For [FP-nets](https://jov.arvojournals.org/ss/deepneuralnets.aspx), we did indeed measure how end-stopped the feature maps of an AND-combination are when presented with a specific input signal. We observed end-stopped activations of different degrees for FP-nets, but only a few end-stopped activations in the baseline architectures. With Min-nets, we have obtained similar results but did not have the space to publish them here.
> Hyper-selectivity directly depends on the angle between the two vectors/filters of a pair. With an angle of 0°, the Min-neuron behaves like an LN-neuron. The Min-neuron becomes more selective when increasing the angle up to 180° – where the neuron only returns zero-values. When considering the angle distributions, the bulk of the values lies within 0° and 100°, with only a few neurons reaching angles up to 160°.
>
> The reviewer has some concerns regarding the biological inspiration and the geometrical interpretation. Hubel and Wiesel have indeed described end-stopping as a neural response that decreases when a bar stimulus is elongated. Although end-stopping has received rather limited attention, there has been some progress regarding the functional role of end-stopping, see, for example, publications such as [Zetsche and Barth](https://pubmed.ncbi.nlm.nih.gov/2392840) and [Zetsche et al.](https://dl.acm.org/doi/10.5555/197765.197778).
> A good starting point regarding the geometric interpretation of both end-stopping and motion selectivity would be [Barth and Watson](https://www.osapublishing.org/oe/fulltext.cfm?uri=oe-7-4-155&id=63497).
> The basic idea is that geometrical flatness (e.g., a straight bar which is stationary or moves uniformly) relates to redundancy and hence neurons - in the spirit of efficient coding - learn to be selective to deviations from flatness.

---

### Decision · Program_Chairs · 2021-11-02

Accept (Poster)

---

> ### Author Response · Authors · 2021-12-03
> **Response to all reviewers**
>
> We thank the reviewers for their constructive criticism and helpful suggestions. Some limitations are due to the limited space that a conference publication offers. However, a related and much more comprehensive paper has been recently accepted for publication in the Journal of Vision ([FP-nets as novel deep networks inspired by vision](https://jov.arvojournals.org/ss/deepneuralnets.aspx)), and some of the issues raised by the reviewers are addressed there; others we would like to address in the following comments.